# Current Knowledge on Mammalian Phospholipase A_1_, Brief History, Structures, Biochemical and Pathophysiological Roles

**DOI:** 10.3390/molecules27082487

**Published:** 2022-04-12

**Authors:** Shun Yaginuma, Hiroki Kawana, Junken Aoki

**Affiliations:** Faculty of Pharmaceutical Sciences, The University of Tokyo, 7-3-1 Hongo, Bunkyo-ku, Tokyo 113, Japan; shunyaginuma-s218@g.ecc.u-tokyo.ac.jp (S.Y.); hiroki-kawana@mol.f.u-tokyo.ac.jp (H.K.)

**Keywords:** phospholipase A_1_, phospholipid metabolism, lysophospholipid, fatty acid

## Abstract

Phospholipase A_1_ (PLA_1_) is an enzyme that cleaves an ester bond at the *sn*-1 position of glycerophospholipids, producing a free fatty acid and a lysophospholipid. PLA_1_ activities have been detected both extracellularly and intracellularly, which are well conserved in higher eukaryotes, including fish and mammals. All extracellular PLA_1_s belong to the lipase family. In addition to PLA_1_ activity, most mammalian extracellular PLA_1_s exhibit lipase activity to hydrolyze triacylglycerol, cleaving the fatty acid and contributing to its absorption into the intestinal tract and tissues. Some extracellular PLA_1_s exhibit PLA_1_ activities specific to phosphatidic acid (PA) or phosphatidylserine (PS) and serve to produce lysophospholipid mediators such as lysophosphatidic acid (LPA) and lysophosphatidylserine (LysoPS). A high level of PLA_1_ activity has been detected in the cytosol fractions, where PA-PLA_1_/DDHD1/iPLA_1_ was responsible for the activity. Many homologs of PA-PLA_1_ and PLA_2_ have been shown to exhibit PLA_1_ activity. Although much has been learned about the pathophysiological roles of PLA_1_ molecules through studies of knockout mice and human genetic diseases, many questions regarding their biochemical properties, including their genuine in vivo substrate, remain elusive.

## 1. Introduction

Phospholipase A_1_ (PLA_1_) is an enzyme that hydrolyzes an ester bond at the *sn*-1 position of glycerophospholipids (GPLs), usually producing a saturated or mono-unsaturated fatty acid and a 1-lyso-2-acyl-phospholipid (2-acyl-lysophospholipid, 2-acyl-LPL) (Figure 1). PLA_1_ has not attracted as much attention as other mammalian acyl hydrolases, such as phospholipase A_2_ (PLA_2_), which hydrolyzes fatty acids, mainly unsaturated fatty acids, and acts as a first step in producing enzymes for bioactive lipids such as eicosanoids and platelet-activating factor (PAF) [1,2]. Some PLA_1_s and PLA_2_ target neutral lipids such as triacylglycerol (TAG) and diacylglycerol (DAG) in addition to GPLs (Figure 1).

Much is known about the functions of PLA_2_, whereas those of PLA_1_ remain limited. However, because fatty acids at both the *sn*-1 and *sn*-2 positions of GPLs have a high turnover rate [3], PLA_1_ as well as PLA_2_, appears to be involved in the rapid turnover and remodeling of cellular GPLs (Figure 2). In addition, some PLA_1_s also have a specific role in the production of 2-acyl-1-lysophospholipids, which serve as lysophospholipid mediators. For example, one type of PLA_1_, membrane-associated phosphatidic acid-selective PLA_1_ (mPA-PLA_1_α in Table 1, Figure 3A), produces 2-acyl-1-lysophosphatidic acid (2-acyl-lysoPA (LPA)) with an unsaturated fatty acid residue [4]. The 2-acyl-LPA acts as a potent ligand for LPAR3/EDG7 and LPAR6/P2Y5, with LPA receptors preferring 2-acyl-LPA over 1-acyl-LPA [5,6]. Phosphatidylserine-specific PLA_1_ (PS-PLA_1_ in Table 1, Figure 3A) also acts as a producing enzyme of another lysophospholipid mediator, 2-acyl-lysophosphatidylserine (2-acyl-lysoPS (LysoPS)), which further supports the idea that PLA_1_s function as producing enzymes for lysophospholipid mediators.

Although PLA_1_ activity has been detected in many mammalian tissues and cells [7,8,9,10,11,12,13,14], only a few PLA_1_s have been purified and cloned. Some triacylglycerol (TAG)-hydrolyzing lipases (Figure 1) such as hepatic lipase, endothelial cell lipase, and lipoprotein lipase also show PLA_1_ activity [15].

## 2. History of PLA_1_ Research

The following is a brief history of mammalian PLA_1_ research. PLA_1_ as well as PLA_2_ activities have been detected in various tissues and plasma. In the 1990’s, two PLA_1_ molecules were biochemically purified and identified. These two PLA_1_s are phosphatidic acid-preferential PLA_1_ (PA-PLA_1_) and phosphatidylserine-specific PLA_1_ (PS-PLA_1_) (Table 1). In 1994, Higgs and Glomset purified a novel PLA_1_ from a cytosolic fraction of bovine testes that preferentially hydrolyzed PA [22] (Table 1). Shortly after, the same group cloned a cDNA of the PLA_1_, and PA-PLA_1_ was found to be an intracellular protein composed of 875-amino acids with a calculated molecular mass of approximately 100 kDa. Simultaneously, Exton’s group purified a very similar PLA_1_ from the bovine brain [37]. It was not clear whether the two intracellular PLA_1_s were identical, because the two groups used different PLA_1_ assay conditions and substrates. Later, Kudo’s group purified similar PLA_1_s from the brain and testes of mice, rats, and cows [38] and showed that they were identical to the PA-PLA_1_ mentioned above. They demonstrated that PA-PLA_1_ was predominantly hydrolyzed phosphatidic acid (PA) in the presence of Triton X-100 and phosphatidylethanolamine (PE) in its absence. The result showed clearly that the substrate specificity of PA-PLA_1_ in vitro is affected by the assay conditions, which makes it difficult to identify the natural substrates of PA-PLA_1_. This also implies that the name, PA-PLA_1_, is not suitable for the enzyme.

Horigome et al., detected two PLA activities in the supernatant of activated rat platelets [39,40]. One was identified as secretory PLA_2_, now known as Group IIA secretory phospholipase A_2_ (sPLA_2_IIA). Sato et al., succeeded in purifying and cloning another PLA that showed a high preference for serine-containing phospholipids [16]. The PLA is now known as PS-specific PLA_1_ (PS-PLA_1_) (Table 1). PS-PLA_1_ specifically hydrolyzed PS in vitro. In addition, it acted on the intact cell membrane, hydrolyzed PS and produced 2-acyl-LysoPS. Thus, PS-PLA_1_ is believed to be a LysoPS-producing enzyme.

The 3D structures of three PLA_1_ family members (extracellular PLA_1_/lipase, cPLA_2_ and PLAAT family members) were shown. For extracellular PLA_1_/lipase family members, β5 and the β9 loops and the lid domain are shown in yellow, cyan and green, respectively. The three residues forming a catalytic triad (Ser, Asp and His) are described as sticks (red). For cPLA_2_ family members (cPLA_2_α, cPLA_2_β and cPLA_2_ζ), the conserved lipase motifs (GXSGX and DXG) are shown in red, and the two residues forming a catalytic dyad (Ser and Asp) are shown as sticks. For PLAAT3/PLA2G16, the three residues forming a catalytic triad (two histidines and cysteine) are shown as sticks (red). The structures without asterisk were acquired from RCSB Protein Data Bank. (Reference PDB IDs; PL (1LPB), LPL (6OB0), PLRP1 (2PPL), cPLA_2_α (1CJY), PLA2G16 (4DOT)). The predicted structures of lipases denoted with an asterisk were acquired from AlphaFold Protein Structure Database. All the structures were visualized using PyMOL software.

After discovering the two PLA_1_ molecules, several researchers found similar PLA_1_s (homologs) in the nucleotide databases, expressed them as recombinant proteins and characterized them biochemically. These analyses identified the homologs as novel PLA_1_s. These include extracellular enzyme membrane-associated PA-selective PLA_1_α (mPA-PLA_1_α, a PS-PLA_1_ homologue) [4] (Table 1) and intracellular enzyme iPLA_1_ γ/DDHD2/KIAA0725 (a PA-PLA_1_ homologue) [24] and iPLA_1_β/SEC23IP/p125 (PLA_1_ activities have not been detected) [27] (Table 1). In addition, similar biochemical characterization was performed for PLA_2_ homologs. Interestingly, biochemical characterizations of the PLA_2_ homologs showed that some of them exhibited PLA_1_ activity in addition to their PLA_2_ activity (PLB activity (Figure 1)) (Table 1). It should also be noted here that certain lipases that hydrolyze TAG have PLA_1_ activity, as mentioned above (Table 1). Lipases hydrolyze fatty acids at the *sn*-1 and *sn*-3 positions of triacylglycerol (TAG) and diacylglycerol (DAG). They also hydrolyze fatty acids at the *sn*-1 or *sn*-3 positions of monoacylglycerol (MAG). This review will summarize current knowledge on PLA_1_ molecules reported thus far, discussing their discoveries, structures, tissue and cellular distributions, and possible biological functions.

PLA_1_s are roughly divided into two groups: (1) extracellular PLA_1_/lipase family and (2) intracellular PLA_1_ family (Table 1), depending on their cellular localization and primary amino acid sequences. The intracellular PLA_1_ family was further subdivided into several groups, consisting of an iPLA_1_ family, PNPLA family, cPLA_2_ family and PLAAT family.

In the following sections, we will summarize what has been known to date about each family member regarding the history of discovery, biochemical characterization, expression, structural feature and function at animal and cellular levels.

## 3. Structural Evaluation of PLA_1_ Molecules

Recently, a computational machine learning method named AlphaFold was developed enabling researchers to predict protein structures with high accuracy, even when no similar experimentally solved structure is available [41]. In this review, we utilized AlphaFold to generate structural predictions of PLA_1_ molecules. Figure 4 summarizes the AlphaFold-generated structures of some PLA_1_ molecules focused on in this review. Note that some structures have been determined, e.g., by X-ray crystallography, while others were structure-predicted. Since structures of any PNPLA and iPLA_1_ family members have not been determined, AlphaFold was unable to predict their structures. Thus, predicted structures were shown only for extracellular PLA_1_/lipase and for cPLA_2_ family members (Figure 4).

## 4. Extracellular PLA_1_/Lipase Family Members

### 4.1. Phosphatidylserine-Specific Phospholipase A_1_ (PS-PLA_1_)

#### 4.1.1. Historical Aspects

PS-PLA_1_ was originally found as a lysophosphatidylserine-selective lysophospholipase (LysoPS-lysophospholipase) that was detected in the cell supernatant of activated rat platelets together with Group IIA secretory phospholipase A_2_ (sPLA_2_ IIA) [40,42]. The purified LysoPS-lysophospholipase was later found also to exhibit PS-PLA_1_ activity [16].

#### 4.1.2. Biochemical Characterization and Tissue Distribution

Activated rat platelets secreted PS-PLA_1_ and sPLA_2_-IIA, both of which are stored in α−granules [40]. Sato et al., purified PS-PLA_1_ from a culture medium of thrombin–activated platelets prepared from approximately 1000 rats [16]. In rats, PS-PLA_1_ is expressed in platelets and in the heart and lung [17,43]. The expression of PS-PLA_1_ in platelets was species-dependent. For example, PS-PLA_1_ mRNA was poorly expressed in human platelets but highly expressed in rat platelets.

Interestingly, enhanced PS-PLA_1_ expression was detected in many tissues when rats were treated with bacterial lipopolysaccharide in vivo (our unpublished observation). Recently, Yatomi et al., established a PS-PLA_1_ immunoassay [44] and subsequently reported that PS-PLA_1_ antigen concentrations were variable in clinical samples from patients. Serum PS-PLA_1_ was significantly elevated in patients with autoimmune disorders, including systemic lupus erythematosus (SLE), rheumatoid arthritis, and Sjogren’s syndrome [45]. Interestingly, the level of PS-PLA_1_ in each SLE individual showed an excellent association with the SLE disease activity index and decreased after the commencement of medical therapy.

PS-PLA_1_ reacts specifically with PS and 1-acyl-LysoPS [39,46]. It cannot appreciably hydrolyze any other phospholipids, including phosphatidyl-D-serine. The enzyme is not a type of PLB because it exclusively hydrolyzes an acyl residue bound at *the sn*-1 position of either PS or 1-acyl-LysoPS. Incubation of the enzyme with 1-acyl-2-radioactive acyl-PS produced radioactive LysoPS but only a tiny amount of radioactive-free fatty acid, even after long-term incubation [16]. It can be concluded that PS-PLA_1_ recognizes a 1-acyl-glycerophospho-L-serine structure and hydrolyzes an acyl residue at *the sn*-1 position. PS-PLA_1_ may first interact with the bilayer structure’s surface to gain access to PS molecules buried in the bilayer.

#### 4.1.3. Structural Characteristics

PS-PLA_1_ belongs to the lipase family, which is composed of classical lipases such as lipoprotein lipase (LPL), hepatic lipase (HL), endothelial lipase (EL), membrane-associated phosphatidic acid-selective PLA_1_α (mPA-PLA_1_α) and mPA-PLA_1_β (Table 1). They share approximately 30–50% amino acid identity. The sequence homology of PS-PLA_1_ was about 80.0% between humans [17] and rats [16]. The deduced amino acid sequence contains a catalytic triad composed of active Ser, Asp, His residues, and “lid” surface loops (Figure 4). Interestingly, conventional lipases have long “lid” (22–23 residues) and “β9” loops (18–19 residues), whereas PS-PLA_1_ has a shorter “lid” (12 residues) and a shorter (deleted) “β9” loop (13 residues) [16,17]. Both loops, which may be strengthened by disulfide bonds linking 14 cysteine residues, have been implicated in substrate recognition. These findings are compatible with the fact that PS-PLA_1_ activity was sensitive to diisopropylfluorophosphate and dithiothreitol [42].

#### 4.1.4. Possible Functions

LysoPS has been shown to stimulate histamine release from rat peritoneal mast cells triggered by cross-linking of FcεRI, a high-affinity receptor for IgE [47,48,49,50]. Interestingly, recombinant PS-PLA_1_ protein had activity similar to that of LysoPS [51], indicating that PS-PLA_1_ interacts with PS in the mast cell plasma membrane to produce LysoPS. Because the PS-PLA_1_-induced degranulation was significantly enhanced in the presence of apoptotic cells [51], cells that surround the mast cells, such as neutrophils in the peritoneal cavity rather than mast cells, must be the source of PS. For example, a crude preparation of rat peritoneal mast cells (containing other cell types such as mononuclear leukocytes and neutrophils) released appreciable amounts of histamine in the presence of PS-PLA_1_ and an Fc receptor cross-linker [51].

PS-PLA_1_ also serves as a LysoPS-producing enzyme for cloned LysoPS receptors (Figure 3A). In a TGFα-shedding assay in which an ectodomain shedding of TGFα detects GPCR activation, PS-PLA_1_ and LysoPS can induce the TGFα shedding in LPSR1/GPR34 (a LysoPS receptor). Thus, it is likely that PS-PLA_1_ acts as a LysoPS-producing enzyme for LPSR1/GPR34. This hypothesis is further supported by the fact that PA-PLA_1_α/LIPH (see below) is a close homolog of PS-PLA_1_ and serves to produce LPA to activate an LPA receptor, LPAR6.

### 4.2. Membrane-Associated Phosphatidic Acid-Selective Phospholipase A_1_s (mPA-PLA_1_α/LIPH and mPA-PLA_1_β)

#### 4.2.1. Historical Aspects

Lysophosphatidic acid (LPA or lysoPA) is a lipid mediator with multiple biological functions. These include physiological functions such as brain development (LPAR1), embryo implantation (LPAR3), blood vessel formation (LPAR4 and LPAR6), and hair follicle development (LPAR6), and has also been implicated in pathological conditions such as the development of fibrosis (LPAR1), endometriosis (LPAR3) and obesity (LPAR4). LPA induces platelet aggregation, smooth muscle contraction, stimulation of cell proliferation, formation of actin stress fibers in fibroblasts, and inhibition of neurite outgrowth in neuronal cells at the cellular level. Recent studies have identified a new family of receptor genes for LPA. A total of three G-protein-coupled receptors (GPCR) belonging to the EDG (endothelial differentiation gene) family (LPAR1/EDG2, LPAR2/EDG4, and LPAR3/EDG7) and another three belonging to the P2Y family (LPAR4/P2Y9, LPAR5/GPR192, LPAR6/P2Y5) have been identified [6,52,53]. LPA with an unsaturated fatty acid at the *sn*-2 position of the glycerol backbone preferentially activates LPAR3 and LPAR6 [5,6]. Interestingly, such LPA with an unsaturated fatty acid may have specific functions. For example, Hayashi et al., found that LPA with an unsaturated fatty acid (unsaturated LPA) but not LPA with a saturated fatty acid stimulated dedifferentiation of rat smooth muscle cells isolated from blood vessels [54]. Tokumura et al., also reported that unsaturated LPA stimulated the proliferation of smooth muscle cells in vitro [55]. Kurano et al., reported that unsaturated LPA such as DHA-LPA and arachidonic acid-containing LPA increased in plasma from acute coronary syndrome (ACS) patients. ACS is the rapid narrowing or occlusion of the coronary artery lumen caused by the collapse of an unstable plaque formed by atherosclerosis and thrombus formation [56].

In the process of the production of unsaturated LPA, PLA_1_ must be involved. There have been two pathways postulated [57]. In one pathway, phospholipids are subjected to the PLA_1_ reaction to produce lysophospholipids with unsaturated fatty acids. Then, the unsaturated lysophospholipids are converted to unsaturated LPA by phospholipase D (PLD) type enzymes. Little is known about the identity of PLA_1_, but the PLD enzyme has been identified as autotaxin (ATX). In the other pathway, phosphatidic acid (PA) is converted to unsaturated LPA by phosphatidic acid-selective phospholipase A1s. Two such PLA_1_s were cloned and characterized, mPA-PLA_1_α [4] and mPA-PLA_1_β [18], both of which are close homologs of PS-PLA_1_ (Table 1). PA-PLA_1_α is expressed explicitly in hair follicles and plays a critical role in forming proper structures of hair follicles by activating LPAR6.

#### 4.2.2. Biochemical Characterization and Distribution

When expressed in insect Sf9 cells, neither mPA-PLA_1_α nor mPA-PLA_1_β protein was recovered from the culture medium, although both have signal sequences at the N-terminus. Instead, both were recovered from the detergent-resistant membrane domains, referred to as raft structures [18].

Human mPA-PLA_1_α is most abundantly expressed in the hair follicle, where, as stated above, it regulates the formation of hair follicles [58]. A closely homologous enzyme, mPA-PLA_1_β, was found in the gene bank. The characteristics of mPA-PLA_1_β were similar to those of mPA-PLA_1_α, except that it was exclusively detected in human testes. Although mPA-PLA_1_β was detected in sperm, only a small amount was detected in seminal fluids [18]. Interestingly, a high mPA-PLA_1_β expression was frequently observed in some rare testes cancers, including Ewing family tumor, although its role in cancer remains unclear [59].

Both mPA-PLA_1_α and mPA-PLA_1_β were found to exhibit PLA_1_ activity against PA. Other GPL substrates and TAG were not substrates for mPA-PLA_1_α and mPA-PLA_1_β [4,18].

#### 4.2.3. Structural Characteristics

The deduced amino acid sequence of mPA-PLA_1_α shares 34.0% identity with that of human PS-PLA_1_ [4], and the N-terminal half of the molecule has about 40% identity with the N-terminal catalytic domain of PS-PLA_1_. In total, three of the amino acid residues in the sequences of mPA-PLA_1_α, Ser-154, Asp-178, and His-248, are completely conserved among lipases (Figure 4). The deduced amino acid sequences of mPA-PLA_1_α and mPA-PLA_1_β are 45.9% identical. The homologous regions were again most prominent in the first half of the molecule. The amino acid lengths of three loop structures, β5, β9 and lid, were exactly the same as in PS-PLA_1_ and mPA-PLA_1_β (Figure 4), while they were quite different in other lipase family members. Thus, these loop structures are believed to be involved in the substrate recognition.

#### 4.2.4. Function

In vitro, both mPA-PLA_1_α and mPA-PLA_1_β showed PA-specific PLA_1_ activity [4,18]. Both enzymes appear to produce LPA at the cellular level. High LPA was detected in cells expressing mPA-PLA_1_α, especially when the cells were treated with bacterial phospholipase D to produce PA on the cell surface [4]. Of note, no appreciable change was observed in the level of any other lysophospholipids under the same conditions [4], indicating that PA-PLA_1_α has high substrate specificity for PA. In 2006, Kazantseva reported that a gene encoding mPA-PLA_1_α (also called Lipase H or LIPH) is a causative gene for an inherited form of hair loss and hair growth defect [60]. After the discovery, similar defects in the mPA-PLA_1_α/LIPH gene were identified all over the world. In addition, a gene whose mutation causes unusual hairiness in dogs and rabbits was reported to encode mPA-PLA_1_α/LIPH. Moreover, we showed that mPA-PLA_1_α/LIPH knockout mice displayed similar woolly or wavy hair [58]. Thus, it is now generally accepted that mPA-PLA_1_α/LIPH plays a critical role in hair growth in a wide range of mammals.

In 2008, Shimomura et al. and Pasternack et al., performed a genetic linkage analysis in several families with autosomal recessive woolly hair and showed linkage to chromosome 13q14.2–14.3 in which an orphan G protein-coupled receptor P2Y5 was mapped [61,62]. Later Yanagida et al., and Inoue et al., showed that P2Y5 was a receptor for LPA, now called LPAR6. Of note, LPAR6 prefers LPA with an unsaturated fatty acid at the *sn*-2 position. Thus, it is reasonable to assume that mPA-PLA_1_α/LIPH is an enzyme that supplies LPA to GPCR-type LPA receptors, also called LPAR6/P2Y5 receptors [63] (Figure 3A).

### 4.3. Extracellular Lipases

In addition to above-described PS- and PA-specific PLA_1_s, specific extracellular lipases exhibit PLA_1_ activity in addition to their intrinsic lipase activity to hydrolyze triacylglycerol (TAG) (Figure 1). These include pancreatic lipase (PL), lipoprotein lipase (LPL), hepatic lipase (HL), endothelial lipase (EL), and pancreatic lipase-related protein 2 (PLRP2) (Table 1), all of which belong to the pancreatic lipase family together with the PS-PLA_1_, mPA-PLA_1_α and mPA-PLA_1_β as mentioned earlier.

#### 4.3.1. Historical Aspects

Lipases hydrolyze TAG present in food or the blood. PL and PLRP2 are secreted from the pancreas and hydrolyze TAG in the small intestine. By contrast, LPL, HL, and EL are mainly present in the blood, where they bind to the surfaces of endothelial cells via heparan sulfate proteoglycans. Both types of lipases catalyze a reaction to hydrolyze ester bonds of TAG at either *sn*-1 or *sn*-3 positions to produce diacylglycerols and fatty acids [15,19]. Interestingly, in addition to their lipase activities, all lipases exhibit considerable PLA_1_ and lysophospholipase activities toward PC and LPC. The precise substrate specificities of the lipases have not been demonstrated, but PC is the most probable phospholipid substrate since its levels are much higher than those of other phospholipid species in both blood and foods.

#### 4.3.2. Structural Characteristics

As noted above, all lipases belong to the pancreatic lipase gene family. The family is conserved in a wide range of animals, from insects to mammals [15]. Presently eight members are known in humans (Table 1). Crystallographic studies of human pancreatic lipases [20] revealed that each of the lipases are composed of N- and C-terminal domains. The N-terminal domains are the sites of catalytic activity. Interestingly, all lipase-like molecules observed in insects (Drosophila melanogaster, Bombyx mori, and hornet) bear only the N-terminal domain [64], supporting the idea that the N-terminal domain is required for catalytic activity. Three amino acid residues in the N-terminal domains, Ser, Asp, and His, form a catalytic triad located in N-terminal domains and are conserved in the pancreatic lipase gene family (Figure 4). In addition, well-conserved cysteine residues that form intramolecular disulfide bonds are present in the N-terminal domains. These structural features indicate that common catalytic machinery is used for both PL and TAG hydrolysis. The crystallographic studies also revealed that human PL contains three surface loops called the lid, the β5 loop, and the β9 loop covering the active sites and have been implicated in the substrate specificity [20,21] (Figure 4). Interestingly, as mentioned above the length of the loop structures are quite different between lipases (PL, LPL, HL and EL) and PLA_1_s (PS-PLA_1_, mPA-PLA_1_α and mPA-PLA_1_β), indicating that these loops determine the substrate specificity of lipase family members.

#### 4.3.3. Functional Characteristics

PL and PLRP2 are present in pancreatic juice and contribute to the intestinal absorption of lipids by hydrolyzing various types of glycerolipids, including TAG, diacylglycerol (DAG), monoacylglycerol (MAG), phospholipids, and lysophospholipids (Figure 3B). The resulting free fatty acids, MAG, and lysophospholipids are then absorbed by intestinal epithelial cells and re-build TAG in the form of lipoproteins (chylomicrons) in the cells.

LPL, HL, and EL share a common role in the metabolism of lipoproteins (Figure 3C). LPL is mainly synthesized by adipose tissues and other tissues such as heart muscle, and HL and EL are synthesized by hepatocytes and endothelial cells, respectively. LPL and HL play roles in the tissue uptake of free fatty acids from lipoproteins. LPL is bound to the capillary endothelium and supplies the underlying tissues with fatty acids derived from the TAG-rich chylomicrons and very-low-density lipoprotein (VLDL). EL and HL exhibit PLA_1_ activities toward PC on HDL in addition to their TAG-lipase activities. Their PLA_1_ activities are partly responsible for their actions in lipoprotein metabolism [65,66]. Interestingly, EL predominantly exhibits PLA_1_ activity, whereas HL exhibits both PLA_1_ and TAG-hydrolyzing activities. Of note, EL inhibition dramatically increases the level of HDL, which suggests that EL PLA_1_ activity has a role in the metabolism of HDL, which is rich in GPLs.

## 5. Intracellular PLA_1_ Families

### 5.1. iPLA_1_ Family

#### 5.1.1. Historical Aspects

Most mammalian cells contain cytosolic PLA_1_s. Cytosolic PLA_1_ activities have been studied in various tissues, including the heart, brain, and testes. Accordingly, a PLA_1_ protein, designated phosphatidic acid-preferring phospholipase A_1_ (PA-PLA_1_, iPLA_1_a) (Table 1), was purified from the brain [37] and testes [22,67]. As with the activities of other lipolytic enzymes, the activities of the PA-PLA_1_s were affected considerably by assay conditions. Using a mixed micelle system, Glomset and colleagues [22] found that a 110 kDa enzyme from testes preferentially hydrolyzed PA, which provides the reason for its name PA-PLA_1_. They cloned a bovine cDNA for PA-PLA_1_ from bovine [23]. PA-PLA_1_ was shown to be a unique phospholipase since it lacked sequence similarity to any phospholipases identified thus far, including extracellular PLA_1_s, lysophospholipase, LCAT, and triacylglycerol lipases. Since PA-PLA_1_ has a DDHD domain, it is also called DDHD1 (Table 1).

Later, two orthologues of PA-PLA_1_ were identified in the database (Table 1). One was p125 which was also identified as a protein that interacted with mammalian Sec23p [27]. Sec23p is a component of the COPII coat that functions in the production of membrane traffic vesicles from the ER [68]. The p125 protein is localized in the ER Golgi intermediate compartment (ERGIC) and cis-Golgi, and its overexpression causes the dispersion of these membrane compartments, suggesting that it is involved in the early secretory pathway. Interestingly, p125 did not exhibit appreciable PLA_1_ activity. The other is KIAA0725p which is also known as DDHD2. KIAA0725p was shown to have PLA_1_ activity for various GPLs, including phosphatidic acid in an assay system containing Triton X-100.

The structures of neither iPLA_1_α/PA-PLA_1_/DDHD1, iPLA_1_β/SEC23IP nor iPLA_1_γ/KIAA0725p/DDHD2 have been determined. Therefore, AlphaFold poorly predicted the precise structures of these proteins, except near the active center, where they displayed weak structural similarities with other hydrolases (Figure 4).

#### 5.1.2. Phosphatidic Acid-Preferring Phospholipase A_1_ (PA-PLA_1_)/DDHD1/iPLA_1_α

##### Characterization and Distribution

PA-PLA_1_ is highly expressed in the testes and brain in a wide range of mammals. It has multiple phosphorylated serine and threonine residues [69,70]. The molecular mass of the expressed PA-PLA_1_ was estimated to be about 110 kDa [22]. The PLA_1_ activity of PA-PLA1 was inhibited by methyl arachidonoyl fluorophosphonate (MAFP), an inhibitor of certain PLA2s, but not by diisopropylfluorophosphate (DFP) [71] a potent inhibitor of PA-PLA_1_. The activity of PA-PLA_1_ was Ca^2+^-independent at a neutral pH but Ca^2+^-dependent at an alkaline pH [71].

##### Substrate Specificity

Studies in vitro showed that the activity of PA-PLA_1_ against PA was 4- to 10-fold greater than the activities of PA-PLA_1_ against PI, PS, PE, and PC in a Triton X-100 mixed micelle system [71]. In the absence of Triton X-100, PE was the best substrate for PA-PLA_1_ [38]. In addition, the PLA_1_ activities were affected by the presence of divalent cations and GPL composition of the substrate [72]. Thus, further studies are needed to identify the real substrates in the cells and in vivo. Precise lipidomic analyses will clarify the substrate in the near future.

##### Function

Spastic paraplegias (SPGs) are neurological disorders characterized by spasticity and gait disturbance. More than 60 types of SPGs caused by mutations in different genes have been reported [73]. SPG type 28 (SPG28) is one type of SPG caused by recessive mutations in the gene encoding PA-PLA_1_α/DDHD1 [74]. How the deficiency of PA-PLA_1_α/DDHD1 leads to the development of the disease has not been identified, but it was shown recently, that the level of lysophosphatidylinositol (LPI) with arachidonic acid (20:4) at the *sn*-2 position decreased in the brains of aged PA-PLA_1_α/DDHD1 knockout mice [75,76], suggesting the potential role of 20:4-LPI in PA-PLA_1_α/DDHD1-mediated signals.

PA-PLA_1_α/DDHD1 is also highly expressed in the testes, where it plays a role in spermatogenesis. Baba et al., reported that PA-PLA_1_α/DDHD1-deficient mice have abnormally shaped sperm, possibly due to the dysregulation of mitochondria of the cells [77]. They speculated that PA metabolism is involved in mitochondrial regulation since PA has been implicated in the fusion and fission cycles of mitochondria. However, one should be cautious about identifying the substrates of PA-PLA_1_/DDHD1, because PA may not be an endogenous substrate for PA-PLA_1_/DDHD1 as discussed above.

#### 5.1.3. KIAA0725p/DDHD2, a Second Member of Intracellular PLA_1_

##### Historical Aspects

A search of the databases revealed a protein encoded by the human expressed sequence tag clone KIAA0725p that exhibited strong sequence similarity (52.6%) to p125 throughout the entire sequence determined [24]. KIAA0725p has 711 amino acids with a calculated molecular mass of about 80 kDa, and similar to p125, it has a Gly-X-Ser-X-Gly consensus sequence. KIAA0725p also contains a DDHD domain and thus has also been called DDHD2 (Table 1).

##### Characterization and Distribution

KIAA0725p is a cytosolic protein. Both Northern and Western blot analyses showed a ubiquitous distribution of KIAA0725p [24]. However, unlike PA-PLA_1_/DDHD1 most of the KIAA0725p/DDHD2 protein was recovered in the membrane fraction. Immunostaining of the protein indicated KIAA0725p/DDHD2 was associated with membrane structures. The post-nuclear supernatant of cells transfected with KIAA0725p cDNA showed high hydrolytic activities against PE and PA and low activities against PS and PC in the absence of Triton X-100. In the presence of Triton X-100, however, it had activity against PA, but only weak activity against PE [24]. KIAA0725p/DDHD2 also exhibits TAG and DAG lipase activities [25,26].

##### Function

Overexpression of KIAA0725p was found to cause a morphological change of organelles, such as dispersion of the ERGIC and Golgi apparatus [24]. Morikawa et al., showed that KIAA0725p is one of the factors mediating a membrane transport pathway between the ER and the Golgi that is distinct from the previously characterized COPI- and Rab6-dependent pathways [78]. KIAA0725p/DDHD2 mutations cause autosomal recessive hereditary spastic paraplegia (SPG54) and abnormal mitochondrial morphology [79]. KIAA0725p/DDHD2 KO mice also show defects in locomotion and cognition, and KO MEF exhibits mitochondrial dysfunction [26,80]. Little is known about the lipids responsible for these abnormalities. Further studies are needed to identify the function of the protein.

### 5.2. Other Recently Identified Intracellular PLA_1_s

After the human genome project was completed, many putative PLA molecules were identified in the databases that showed significant homology to previously identified and characterized PLA_1_/PLA_2_ molecules. Of note, some molecules were indeed shown to display PLA_1_, lysophospholipase, and phospholipase B activities. An example is PLB, which hydrolyzes a fatty acid at both *sn*-1 and *sn*-2 positions. Here, we summarize the molecular features of these enzymes.

#### 5.2.1. iPLA_2_/PNPLA Family

The iPLA_2_ (Ca^2+^-independent phospholipase A_2_) family members adopt a three-layer α/β/α architecture and harbor an enzymatically active site composed of a Ser-Asp catalytic dyad. They are also called PNPLA (patatin-like phospholipase domain-containing) family members [81] because they do not require Ca^2+^ ions for their activity and have a domain structure similar to that of plant lipase patatin family members. Some of these enzymes exhibit not only PLA_2_ activity but also PLA_1_ and lysophospholipase activity (PLB (phospholipase B) activity). Among the eight members of the iPLA_2_ family, at least three of them (PNPLA6/iPLA_2_δ/NTE, PNPLA7/iPLA_2_θ/NRE, and PNPLA8/iPLA_2_γ/group VIB PLA_2_) (Table 1) appear to have PLA_1_ activity.

##### PNPLA6/iPLA_2_δ/NTE

Historical Aspects

PNPLA6 is also known as NTE (neural target esterase) since it was identified as a target molecule for organophosphorus insecticides, which cause neurological disorders such as paralysis of the lower limbs due to degeneration of axons in the spinal cord and peripheral nerves [82]. Later the endogenous substrates for this enzyme were suggested to be GPLs, as shown in the following sections.

Biochemical Characterization and Tissue Distribution

PNPLA6 shows lysophospholipase activity toward lysophospholipids (especially LPC), producing fatty acids and glycerophosphocholine (GPC) [28]. The enzyme is also conserved in yeast and fruit flies, and these orthologues show phospholipase B activity, hydrolyzing fatty acids both at the *sn*-1 and *sn*-2 positions of PC to produce LPC and GPC [29,30]. PNPLA6 is highly expressed in the nervous system.

Structural Characteristics

PNPLA6 contains an extended N-terminal domain and three nucleotide-binding motifs in contrast to the common Patatin domain of the iPLA_2_ family. PNPLA6 has a transmembrane region near the N-terminus, which was not found in other family members except for PLPLA7.

Possible Functions

Systemic homozygous deficiency of PNPLA6 in mice has been reported to result in fetal lethality due to placental dysfunction. Heterozygous mice show an increased spontaneous locomotor activity and hyperactivity tendencies [82,83]. Progressive neurodegenerative abnormalities were observed in the hippocampus, thalamus, and cerebellum in mice with neuron-specific deficiency [84]. Mutations near the active center of PNPLA6 in humans have been reported to be responsible for hereditary diseases such as spasticity, ataxia, hypogonadism, and retinal degeneration (Boucher–Neuhäuser syndrome, Laurence–Moon syndrome, Oliver–McFarlane syndrome, and spastic paraplegia) [85,86,87]. The molecular mechanism underlying the PNPLA6-deficiency-induced neurodegeneration remains obscure. However, it has been proposed that glycerophosphocholine (GPC), the final product of PLB reaction from PC, produced in the kidney, may function as an osmotic regulator since its expression is induced by high NaCl concentrations [88].

##### PNPLA7/iPLA2θ/NRE

Historical Aspects

PNPLA7 is a close relative of PNPLA6/NTE and is therefore named NRE (NTE-related esterase).

Biochemical Characterization and Tissue Distribution

Similar to PNPLA6, PNPLA7 shows lysophospholipase activity for LPC, producing fatty acids and GPC [31]. While PNPLA6 is predominantly expressed in the nervous system, PNPLA7 is mainly found in peripheral tissues such as the testes, skeletal muscle, heart, and adipose tissue.

Structural Characteristics

PNPLA7 is a close homolog of PNPLA6 [31,89] and has a similar domain structure to PNPLA6, with an N-terminal transmembrane region and three nucleotide-binding motifs [31,89].

Possible Functions

PNPLA7 is involved in choline metabolism by hydrolyzing LPC [31]. Interestingly, PNPLA7 is localized to the ER and tiny lipid droplets (LDs) in a catalytic domain-dependent manner [90]. PNPLA7-deficient mice were recently shown to have a premature aging phenotype and perish before adulthood due to metabolic defects, particularly in the liver (Makoto Murakami personal communication). However, the underlying molecular mechanisms remain unclear.

##### PNPLA8/iPLA_2_γ/group VIB PLA_2_

Historical Aspects

PNPLA8 is synonymous with a Ca^2+^-independent PLA_2_γ (iPLA_2_γ) and was originally cloned from human heart cDNA [91].

Biochemical Characterization and Tissue Distribution

PNPLA8/iPLA_2_γ exhibits a unique enzymatic activity [91]. It hydrolyzes a fatty acid at the *sn*-2 position (PLA_2_ activity) when alkenyl PC is used as a substrate in vitro. It also hydrolyzes a fatty acid at the *sn*-1 position (PLA_1_ activity) when diacyl PC with a polyunsaturated fatty acid at the *sn*-2 position is used as a substrate [32]. In humans, PNPLA8/iPLA_2_γ mRNA expression is exceptionally high in the heart [91,92]. Immunohistochemical analysis in mouse myocardium demonstrated that PNPLA8/iPLA_2_γ is associated with mitochondria and peroxisomes, reflecting dual protein localization motifs.

Structural Characteristics

PNPLA8 is a close homolog of PNPLA9/iPLA_2_β and has a similar active center but less homology at the N-terminus [93]. PNPLA9/iPLA_2_β has four possible translation initiation sites, giving rise to proteins of 88, 77, 74, and 63 kDa. The N- and C-termini have mitochondrial and peroxisomal localization sequences, respectively, and the full-length 88- and 63 kDa translation products were observed to localize to mitochondria and peroxisomes, respectively, in the cell [94,95].

Possible Functions

Liu et al., suggested that PNPLA8 is involved in cardiolipin remodeling and is essential for maintaining mitochondrial function [96]. A variety of phenotypes have been reported in PNPLA8-deficient mice [33], including failure to thrive, reduced resistance to exercise and myocardial stress, muscle weakness, decreased body temperature, adipose tissue atrophy, and neurodegeneration. Mutations in PNPLA8 in humans are associated with mitochondrial myopathy [97]. In a mouse model of myocardial ischemia, the levels of potential lipid mediators, including eicosanoids and lysophospholipids were altered in PNPLA8-deficient and PNPLA8-overexpressing mice, suggesting that PNPLA8 also has a function in the production of these lipid mediators [98]. In the liver, PNPLA8 is a candidate enzyme that supplies LPC, the substrate of PNPLA7.

#### 5.2.2. cPLA_2_ Family

The Group IV PLA_2_ family is comprised of six intracellular enzymes commonly called cytosolic PLA_2_α (cPLA_2_α), cPLA_2_β, cPLA_2_γ, cPLA_2_δ, cPLA_2_ε, and cPLA_2_ζ. Interestingly, they are mostly homologous to PLA and PLB/lysophospholipase of filamentous fungi, particularly in regions containing conserved residues involved in catalysis. Their primary functions appear to be exerted through their PLA_2_ activity providing arachidonic acid for synthesis of eicosanoids. However, some members, i.e., cPLA_2_α/PLA_2_G4A/GroupIVA PLA_2_, cPLA_2_β/PLA_2_G4B/GroupIVB PLA_2_, and cPLA_2_ζ/PLA_2_G4F/Group IVF PLA_2_ also exhibit PLA_1_ activity [34,35]. It should be noted that all these cPLA_2_s have their own abilities to carry out multiple reactions to varying degrees (PLA_2_, PLA_1_, lysophospholipase and transacylase activities). Thus, the conclusion that an enzyme has PLA_1_ activity must be evaluated carefully, as an enzyme with both PLA_2_ and lysophospholipase activity will apparently be judged to exhibit PLA_1_ activity in an in vitro assay. The function of the PLA_1_ activity of these enzymes remains completely unknown.

The X-ray crystal structures of cPLA_2_ family members were determined only for cPLA_2_α [99] (Figure 4). The structure explained the selectivity of the enzyme with arachidonic acid-containing GPLs using an unusual Ser-Asp dyad located in a deep cleft at the center of a predominantly hydrophobic funnel. AlphaFold predicted similar structures for cPLA_2_β and cPLA_2_ζ (Figure 4).

##### PLA2G16/group XVI PLA_2_/PLAAT3/HRASLS3/H-Rev107

Historical Aspects

PLAAT3 was found by subtraction cloning between sensitive and resistant fibroblasts to transformation by introducing H-Ras [100]. Overexpression of PLAAT3 in H-Ras-transformed fibroblasts suppressed cell proliferation and colony formation, indicating that PLAAT3 negatively regulates H-Ras’ function [101].

Biochemical Characterization and Tissue Distribution

PLAAT3 hydrolyses fatty acids at the *sn*-1 or *sn*-2 positions of PC and PE. Thus, it has both PLA_1_ and PLA_2_ activities, although the former is stronger than the latter. In vitro, PLAAT3 also shows weak N-acyltransferase and O-acyltransferase activities compared to other PLAAT family members [36,102]. PLAAT3 is highly expressed in adipocytes and is also known as adipose-specific PLA_2_ (Ad-PLA_2_) [103].

Structural Characteristics

PLAAT3 has homology to lecithin-retinol acyltransferase (LRAT) and belongs to the NlpC/P60 superfamily. PLAAT3 contains a histidine-containing three-amino acid residue called the H-box and a cysteine-containing domain called the NC domain, which is thought to form the active site. In addition, it bears a proline-rich domain at the N-terminal side and a membrane-binding domain with a cluster of hydrophobic amino acids at the C-terminal side. These domains were reported to be essential for the regulation of H-Ras function [101]. A recent crystal structure analysis of PLAAT3 (Figure 4) revealed the catalytic mechanism of the enzyme [104].

Possible Functions

PLAAT3-deficient mice are resistant to diet-induced obesity [105]. PLAAT3 is also involved in cancer invasion and metastasis [106] and is known to be involved in vitamin A metabolism [107], promotion of peroxisome formation, and production of ether GPLs [108]. Recently, Morishita et al., reported that PLAAT3 was involved in organelle degradation in the eye lens [109]. The eye lens of vertebrates is composed of fiber cells in which all membrane-bound organelles, including mitochondria, the endoplasmic reticulum, and lysosomes, undergo degradation during terminal differentiation to form an organelle-free zone. PLAAT3 in mammals and Plaat1 (functional homolog) in zebrafish were shown to be essential in the organelle degradation in the eye lens.

Interestingly, these enzymes translocate from the cytosol to various organelles immediately before organelle degradation. The C-terminal transmembrane domain of the enzymes was shown to be essential to the process. It is speculated that for the complete digestion of GPLs, both PLA_1_ and PLA_2_ activities (PLB activity) of PLAAT3 are needed.

## 6. Conclusions and Perspectives

In this article, we summarize our current understanding of PLA_1_ molecular structures and its activities. Genome projects have revealed a wide range of PLA_1_-like molecules in the human genome, and subsequent biochemical studies have also revealed their nature. Of note, extracellular PLA_1_ molecules such as mPA-PLA_1_α and PS-PLA_1_ are now more clearly accepted as enzymes that produce lysophospholipid mediators such as LPA and LysoPS. Intracellular PLA_1_s were also identified and an analysis of their mutants revealed that they exhibited essential physiological functions. However, the precise substrates and products and the physiological significance of the enzymatic reactions remain to be elucidated. Comprehensive lipidomics analysis of PLA_1_ mutants and their overexpressing cells will help elucidate the true substrates, products, and functions of PLA_1_ in future studies.

## Figures and Tables

**Figure 1 molecules-27-02487-f001:**
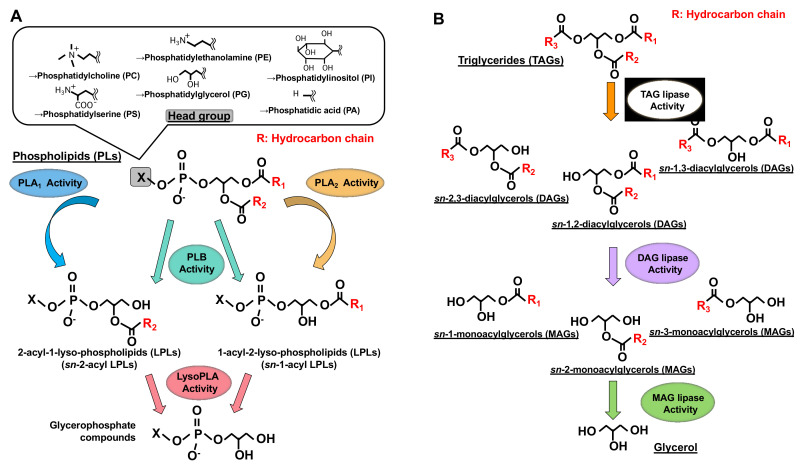
Structures of glycerolipids and their metabolic enzymes. (**A**) Glycerophospholipids (GPL) and phospholipases. GPLs are composed of a polar head group (six major classes), a glycerol backbone and fatty acid moieties (esterified at the *sn*-1 and *sn*-2 positions). Phospholipase A_1_ (PLA_1_) hydrolyzes a fatty acid at the *sn*-1 position, generating *sn*-2-acyl-1-lyso-phospholipids (*sn*-2-acyl LPLs), while phospholipase A_2_ (PLA_2_) hydrolyzes a fatty acid at the *sn*-2 position generating *sn*-1-acyl-2-lyso-phospholipids (*sn*-1-acyl LPLs). Phospholipase B (PLB) hydrolyzes a fatty acid at both *sn*-1 and *sn*-2 positions. LysoPLA hydrolyzes a fatty acid of *sn*-2-acyl LPLs and *sn*-1-acyl LPLs, generating glycerophosphate compounds. (**B**) Triacylglycerol (TAG) has three fatty acids at the *sn*-1, *sn*-2 and *sn*-3 positions of glycerol backbone, diacylglycerol (DAG) has two fatty acids and monoacylglycerol (MAG) has one fatty acid. TAG lipase hydrolyzes a fatty acid of TAG, generating *sn*-1, 2, *sn*-2, 3 or *sn*-1, 3-diacylglycerols (DAGs). DAG lipase hydrolyzes a fatty acid of DAG and MAG lipase hydrolyzes a fatty acid of MAG.

**Figure 2 molecules-27-02487-f002:**
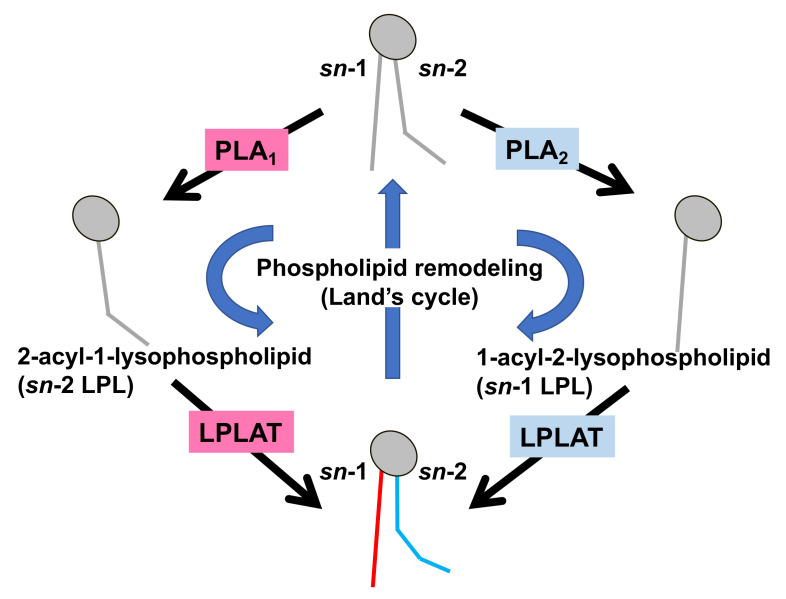
Fatty acid remodeling reactions of GPLs. Glycerophospholipids (GPL) in the cells are constantly subjected to two kinds of fatty acid hydrolyzing reactions mediated by phospholipase A_1_ (PLA_1_) and phospholipase A_2_ (PLA_2_), resulting in the production of 2-acyl-1-lysophospholipid (*sn*-2 LPL) and 1-acyl-2-lysophospholipid (*sn*-1 LPL). The LPLs thus produced are further subjected to acylation reactions to re-form the GPLs. Several kinds of lysophospholipid acyltransferases (LPLAT) are responsible for the introduction of fatty acids to lysophospholipids. By these sequential GPL remodeling reactions, the fatty acids of GPLs are constantly replaced.

**Figure 3 molecules-27-02487-f003:**
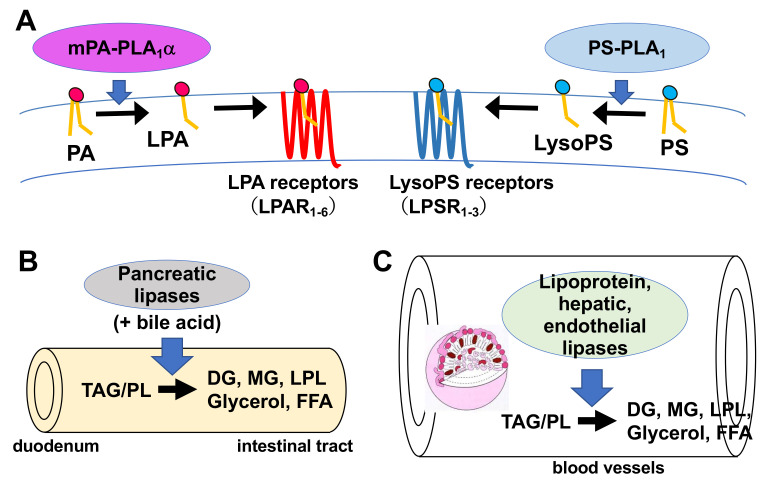
Physiological roles of extracellular PLA_1_s. (**A**) PS-PLA_1_ and mPA-PLA_1_α serve as producing enzymes for lysophospholipid mediators. PS-PLA_1_ has a strict substrate specificity in that it only acts on serine containing GPLs such as phosphatidylserine (PS) and lysophosphatidylserine (LysoPS). LysoPS then acts on GPCR-type LysoPS receptors. Three such LysoPS receptors have been identified. These include LPSR1/GPR34, LPSR2/P2Y10, and LPSR3/GPR174. mPA-PLA_1_α acts on PA in a specific manner and produces *sn*-2 LPA, which then acts on GPCR-type LPA receptors, LPAR1-LPAR6 evoking various biological responses. (**B**) Pancreatic lipase (PL) is secreted from the pancreas into the lumen of the intestine, where it, with the aid of bile acids, hydrolyzes the fatty acids of triacylglycerol (TAG) and GPLs in the digestive juice yielding diacylglycerol (DAG), monoacylglycerol (MAG) and fatty acids. The liberated fatty acids are absorbed by intestinal cells as nutrients. (**C**) Lipoprotein lipase (LPL), hepatic lipase (HL), and endothelial lipase (EL), which are mainly present in the blood, are associated with endothelial cell surfaces in adipose tissues (LPL), heart (LPL), liver (HL) and various tissues. These lipases have both TAG lipase and PLA_1_ activities. They hydrolyze fatty acids of TAG and GPLs present in the circulating lipoproteins such as low-density lipoproteins (LDL) and high-density lipoproteins (HDL), yielding diacylglycerol (DAG), monoacylglycerol (MAG), lysophospholipids (LPL) and fatty acids. The free fatty acids are absorbed by corresponding cells for energy source and storage of neutral lipids.

**Figure 4 molecules-27-02487-f004:**
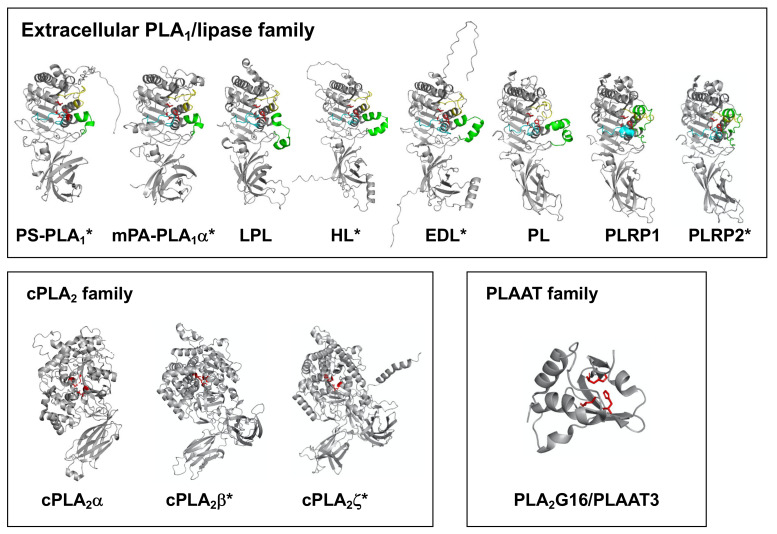
Structures of PLA_1_ molecules. 3D structures of three PLA1 family members (extracellular PLA1/lipase, cPLA2 and PLAAT family members) were shown. For extracellular PLA1/lipase family members, β5 and the β9 loops and the lid domain are shown in yellow, cyan and green, respectively. The three residues forming a catalytic triad (Ser, Asp and His) are described as sticks (red). For cPLA2 family members (cPLA2α, cPLA2β and cPLA2ζ), the conserved lipase motifs (GXSGX and DXG) are shown in red, and the two residues forming a catalytic dyad (Ser and Asp) are shown as sticks. For PLAAT3/PLA2G16, the three residues forming a catalytic triad (two histidines and cysteine) are shown as sticks (red). The structures without asterisk were acquired from RCSB Protein Data Bank. (Reference PDB IDs; PL (1LPB), LPL (6OB0), PLRP1 (2PPL), cPLA2α (1CJY), PLA2G16 (4DOT)). The predicted structures of lipases with asterisk were acquired from AlphaFold Protein Structure Database. All the structures were visualized using PyMOL software.

**Table 1 molecules-27-02487-t001:** Mammalian PLA_1_s.

	Primary Name	Other Name	Human Gene	Substrate	Reaction Mediated	PDB ID (H: Human, R: Rat)	Ref.
extracellular PLA_1_s	PS-PLA_1_	PLA1A	*PLA1A*	PS	Producing enzyme for bioactive lysophospholipid, LysoPS	-	[16,17]
	PA-PLA_1_α	LIPH, mPA-PLA_1_α	*LIPH*	PA	Producing enzyme for bioactive lysophospholipid, LPA	-	[4,18]
	lipoprotein lipase	LPL, LIPD	*LIPD*	TAG, PL	TAG lipase and PLA_1_ activity	H: 6E7K, 6OAU, 6OAZ, 6OB0, 6WN4	[19]
	hepatic lipase	HL, LIPC	*LIPC*	TAG, PL	TAG lipase and PLA_1_ activity	-	[19]
	endothelial cell-derived lipase	EDL, EL, LIPG	*LIPG*	PL	Predominant PLA_1_ activity	-	
	pacreatic lipase	PL, PNLIP	*PNLIP*	TAG, PL	TAG lipase and PLA_1_ activity	H: 1GPL, 1LPA, 1LPB, 1N8S	[15,20,21]
	pancreatic lipase-related protein 1	PLRP1	*PLRP1*	TAG, PL	TAG lipase and PLA_1_ activity	H: 2PPL	
	pancreatic lipase-related protein 2	PLRP2	*PLRP2*	TAG, PL	TAG lipase and PLA_1_ activity	H: 2OXE, 2PVS; R: 1BU8	[15,21]
intracellular PLA_1_s	PA-PLA_1_	DDHD1, iPLA_1_α	*DDHD1*	PL	PLA_1_ activity	-	[22,23]
	KIAA0725p	DDHD2, iPLA_1_γ	*DDHD2*	PL	PE, DAG, CL	-	[24,25,26]
	p125	iPLA_1_β	*P125*	n.d.	Enzymatic activity has not been detected	-	[27]
	PNPLA6	iPLA_2_δ, NTE	*PNPLA6*	PC, LPC	PLB, LysoPLA activity cleaving FAs at both *sn*-1 and *sn*-2 positions	-	[28,29,30]
	PNPLA7	iPLA_2_θ, NRE	*PNPLA7*	PC, LPC	PLB, LysoPLA activity cleaving FAs at both *sn*-1 and *sn*-2 positions	-	[31]
	PNPLA8	iPLA_2_γ, Group VIB PLA_2_	*PNPLA8*	PC	PLB activity cleaving FAs at both *sn*-1 and *sn*-2 positions	-	[32,33]
	cPLA_2_α	PLA2G4A, Group IVA PLA_2_	*PLA2G4A*	PL	PLB activity cleaving FAs at both *sn*-1 and *sn*-2 positions	H: 1BCI, 1CJY, 1RLW	[34]
	cPLA_2_β	PLA2G4B, Group IVB PLA_2_	*PLA2G4B*	PL	PLB activity cleaving FAs at both *sn*-1 and *sn*-2 positions	-	[34,35]
	cPLA_2_ζ	PLA2G4F, Group IVF PLA_2_	*PLA2G4F*	PL	PLB activity cleaving FAs at both *sn*-1 and *sn*-2 positions	-	
	PLA2G16	Group XVI PLA_2_, PLAAT3, HRASLS3, H-Rev107	*PLA2G16*	PL	PLB activity cleaving FAs at both *sn*-1 and *sn*-2 positions	H: 2KYT, 4DOT, 4FA0, 4Q95, 7C3Z, 7C41	[36]

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
