# Peer review of "Current Knowledge on Mammalian Phospholipase A_1_, Brief History, Structures, Biochemical and Pathophysiological Roles"

_molecules, 2022, doi:10.3390/molecules27082487_

Round 1

Reviewer 1 Report

In this manuscript, Junken Aoki et al presented the history of the research of phospholipase A1 and summarized the discovery, biochemical characterization, and pathology of the PA and PS specific-PLA1s. In general, this work provides a systemic review and up-to-date introduction to the field and I consider this as a suitable article for publication in the Molecules. There are still several concerns to be addressed and I hope the authors could polish their text and figure before publication.

1) The graphic illustration in this manuscript needs significant polishing and improvement to deliver the author’s thoughts in a clearer way. For example, in Figure 2, sn1 and sn2 should be labeled for lipid models. In Figure 1, the author should at least re-draw the cartoon representation for the bilayer membrane since it looks the lipid “PA”, “LPA” and “PS” are much bigger than the membrane itself. In addition, in line 47, Figure 1 legend, mPA-PLA1a should be written as mPA-PLA1α.
2) The authors use “LysoPS” for lysophosphatidylserine and “LPA” for lysophosphatidic acid. I will suggest a universal name system used in one manuscript and probably lysoPS and lysoPA might be better.
3) Also for the lysolipid receptors, such as receptors for LPA and LPS, authors used confusing protein names, LPA1-6 and LPS1-3, which makes the readers spend a lot of time distinguishing those GPCRs from their ligands since they are under the same names. I will suggest that the official name LPAR1-6 and LPSR1-3 should be used for those receptors.
4) There is no citation in line 207-218 and if the data and information are from other articles, please cite them. If this is unpublished data, the author should indicate it.
5) Authors should check their grammar and typos. For example, P9-L367,”The post-nuclear supernatant of cells transfected with KIAA0725p cDNA showed high hydrolytic activities against PE and PA and low activities against PS and PC in the absence of Triton X-100 presence of Triton X-100”.

In summary, my major concern is about the terms and abbreviations used in this manuscript and the author should make sure of consistency. Adding an abbreviation table at the beginning of the paper might be helpful.

Reviewer 2 Report

The manuscript molecules-1471739 by Junken Aoki et al. entitled “Current knowledge on mammalian phospholipase A1-History, structures, biochemical and pathophysiological roles –“reviews the history and present knowledge of phospholipases A1 (PLA1s). This comprehensive review compares the history, biochemical characterization, structure, substrate specificity and functions of  different families of PLA1s. The review is well written and it offers a solid overview for any reader interested in PLA1s. Thus, It is a review worth publishing in molecules.

Minor comments:

The different subsections (historical aspects, biochemical characterization and tissue distribution, structural characteristics, substrate specificity and possible functions) should be the same for all the families). If this is not possible, some subsections could be combined.

The information provided in the subsections “structural characteristics” should be complemented with a figure containing a representation of the different structures described in the text. Authors should provide the pdb ID of these structures.

It would be good to include a figure with the molecular representation of the different substrates recognized and hydrolyzed by the different PLA1s families.

Fig.2 (line 35) is mentioned in the main text before Fig.1A (line 37)

Fig. 4 mentioned in line 158 is not present in the manuscript.

Reviewer 3 Report

This manuscript is a sequential review of reports on mammalian phospholipases.
It is expected that this manuscript will be helpful in understanding the comprehensive research trend of various mammalian phospholipases.
However, there are parts of the written manuscript that have low readability or poor understanding. In order for these manuscripts to be published in the journal, the following items need to be improved.

  1. The author mentioned history and structure in the title, but the proportion in the manuscript is very small.
    Most of the history is only a few lines (historical aspects), and most of it is all that has been found.
    In the case of structure, there is not enough content to understand with text alone without structure figures.
    In the case of history, it is appropriate to supplement the content or to exclude it from the title, and in the case of structure, it is suggested to make the related structure into a figure.
  2. Figure 1 and Figure 2 are of very low quality to be published on Journal. Also, at the bottom of the figure, “Yaginuma et al. Fig. 1” should be removed.
  3. Line 158: (see the upper-right cell in Fig. 4)." There is no Fig. 4 in the manuscript.
  4. Line 34: (Fig. 2) precedes Figure 1.
  5. Line 73: "Dr. Glomset and his colleagues" should be "Higgs and Glomset" or "Glomset and his colleagues"
  6. Authors should use consistent expressions.
    e.g.
    Line 328: Ca2+-independent
    Line 328: Ca2+-dependent
    Line 390: calcium-independent
    Line 451: calcium-independent
  7. e.g. 
    Line 766; 97,576 daltons (Da)
    Line 307: 110-kDa
    Line 325: 110 KDa
    Line 359: 81,003 Da
  8. References should be made clear in the sentences below.
    -Line 238-239: "We also showed that PA-PLA1/LIPH knockout mice showed similar woolly or wavy hair. Thus, now it is generally accepted that PA-PLA1/LIPH has a critical role in hair growth in a wide range of mammals."
    -Line 440-441: "PNPLA7 is a close homolog of PNPLA6 and has a similar domain structure to PNPLA6, with an N-terminal transmembrane region and three nucleotide-binding motifs."
    -Line 451: "PNPLA8 is synonymous with a calcium-independent PLA2 (iPLA2) and was originally cloned from a human heart cDNA."
    -Line 457-460: "In humans, PNPLA8/iPLA2 mRNA expression is exceptionally high in the heart. Immunohistochemical analysis in mouse myocardium demonstrated that PNPLA8/iPLA2 is associated with mitochondria and peroxisomes, reflecting dual protein localization motifs."

Minor 
-In text, "(Table)" should be "(Table 1)"
-line 391: "a/b/a architecture" should be "α/β/α architecture"

Round 2

Reviewer 3 Report

The authors have addressed all of my concerns.  I support the publication of this revised manuscript on Molecules.